# Incompleteness of Electronic Health Records: An Impending Process Problem Within Healthcare

**DOI:** 10.3390/healthcare13222900

**Published:** 2025-11-13

**Authors:** Varadraj Gurupur, Sahar Hooshmand, Deepa Fernandes Prabhu, Elizabeth Trader, Sanket Salvi

**Affiliations:** 1Center for Decision Support Systems and Informatics, University of Central Florida, Orlando, FL 32816, USA; deepa.fernandesprabhu@ucf.edu (D.F.P.); elizabeth.trader@ucf.edu (E.T.); sanket.salvi@ucf.edu (S.S.); 2School of Global Health Management and Informatics, University of Central Florida, Orlando, FL 32816, USA; 3Department of Computer Science, California State University, Dominguez Hills, Carson, CA 90747, USA; hooshmand@csudh.edu; 4School of Modeling, Simulation and Training, University of Central Florida, Orlando, FL 32816, USA; 5Department of Electrical and Computer Engineering, University of Central Florida, Orlando, FL 32816, USA

**Keywords:** electronic health records, incompleteness, data quality, missing data, interoperability, bias, clinical documentation, machine learning, process improvement, global health

## Abstract

Background: The digitization of health records was expected to improve data quality and accessibility, yet incompleteness remains a widespread challenge that undermines clinical care, interoperability, and downstream analytics. Problem: Evidence shows that missing and under-recorded elements in electronic health records (EHRs) are largely driven by process gaps across patients, providers, technology, and policy—not solely by technical limitations. Objective: This perspective integrates conceptual foundations of incompleteness, synthesizes cross-country evidence, and examines process-level drivers and consequences, with an emphasis on how missingness propagates bias in AI and machine learning systems. Contribution: We present a unifying taxonomy, highlight complementary approaches (e.g., Record Strength Score, distributional testing, and workflow studies), and we propose a pragmatic agenda for mitigation through technical, organizational, governance, and patient-centered levers. Conclusions: While EHR incompleteness cannot be fully eliminated, it can be systematically mitigated through standards, workflow redesign, patient engagement, and governance—essential steps toward building safe, equitable, and effective learning health systems.

## 1. Introduction

Electronic health records (EHRs) have become a cornerstone of modern healthcare, providing critical infrastructure for clinical decision-making, research, and health system management. Global adoption has grown rapidly: many high-income countries now have national or regional EHR systems in place (often exceeding 50–70% adoption), whereas adoption in low- and lower-middle-income countries remains substantially lower, often in the range of 10–40%, depending on region [1]. In the United States, nearly 96% of hospitals and 78% of office-based physicians now use certified EHR systems [2]. Similarly, the European Union has committed to creating a European Health Data Space by 2025, aiming to improve cross-border interoperability of EHRs [3].

Despite this widespread implementation, the incompleteness of EHR data remains a critical, unresolved challenge. Empirical studies reveal that many EHR datasets have substantial missingness: in some settings, 30–40% of variables may be missing more than half their expected values, especially for laboratory or social determinants data, and overall missingness rates for key clinical variables often fall in the 15–30% range, depending on context [4,5,6]. Missing or fragmented EHR records have been linked to an increased risk of medication errors, redundant testing, misdiagnoses, and worse health outcomes, especially among marginalized populations. Evidence shows that discontinuities in patient records degrade predictive model fairness for racial and ethnic minorities [7], while missing data and documentation gaps contribute to diagnostic inaccuracies and bias in clinical decision-making [8,9]. As health systems increasingly rely on EHR data for predictive analytics, population health management, and machine learning applications, the risks associated with incompleteness are magnified.

While prior research has described aspects of data incompleteness, most existing work remains fragmented. Earlier contributions have focused either on conceptual frameworks [10], technical completeness metrics [11], or specific methodological biases [12]. However, a unifying perspective that situates incompleteness as a systemic process problem spanning patients, providers, technology, and policy has been largely absent. Furthermore, the global dimension of this issue manifested in differences across health systems, governance structures, and digital divides has not been adequately integrated into prior reviews. The novelty of this article lies in the following contributions:It provides an integrative framing of EHR incompleteness across six interdependent domains: definitions and dimensions, types of missingness, research contributions, process gaps, implications, and mitigation strategies with future directions.It synthesizes empirical findings from multiple countries, moving beyond single-region perspectives.It highlights both the statistical foundations and the sociotechnical drivers of incompleteness.It emphasizes the ethical and policy consequences of incomplete records, linking data quality to equity and governance.

In parallel with these academic and operational developments, major strategic and regulatory bodies have also advanced initiatives to improve the completeness, interoperability, and governance of electronic health records. For instance, the European Commission’s European Health Data Space initiative seeks to harmonize standards and enhance the quality and accessibility of health data across member states, promoting cross-border interoperability and secondary data use for research and innovation [3,13,14]. Similarly, the U.S. Food and Drug Administration (FDA) has emphasized real-world data reliability and electronic health record usability as part of its broader digital health modernization efforts, reinforcing the importance of structured and complete data capture for regulatory decision-making. These activities reflect growing international momentum toward policy frameworks that prioritize data completeness as a foundation for safe, equitable, and interoperable digital health systems [13,15].

### 1.1. Perspective Design and Scope

As this work is presented as a perspective, rather than a systematic review, its objective is to synthesize key conceptual, empirical, and policy-oriented developments related to EHR incompleteness. The selection of literature emphasized peer-reviewed studies published between 2013 and 2025 that addressed data completeness, interoperability, bias, and governance within electronic health record systems. Foundational theoretical frameworks were included to establish conceptual grounding, complemented by recent computational and policy-driven studies that reflect current trends. The perspective was designed to integrate diverse viewpoints—spanning technical, organizational, patient-centered, and policy domains—into a coherent argument highlighting incompleteness as a systemic process issue. This narrative and integrative design was chosen to encourage cross-disciplinary dialogue and to identify pragmatic directions for future research and governance.

The remainder of this article is structured as follows: Section 2 develops the conceptual foundations of incompleteness, including definitions, mathematical representations, and a taxonomy of missingness. Section 3 reviews key research contributions, bridging conceptual, statistical, and computational perspectives. Section 4 synthesizes cross-country evidence, underscoring the global scope of the problem. Section 5 identifies process gaps across patients, providers, technology, and policy. Section 6 analyzes the implications of incompleteness for clinical care, operations, research, and ethics. Section 7 outlines mitigation strategies and future directions, spanning technical, organizational, policy, and patient-centered solutions. Finally, Section 8 presents conclusions and authorial reflections.

### 1.2. Definitions and Dimensions

The IEEE Standards Association defines incompleteness in electronic health records (EHRs) as the absence of a required data element that should exist for a patient encounter [16]. Building upon this, Weiskopf et al. [10] proposed four operational dimensions:Documentation—whether a data element is recorded at all.Breadth—the extent to which all relevant data types are captured.Density—the frequency of data entries over time.Predictive value—the ability of a record to support reliable inference about outcomes.

As shown in Table 1, each dimension of completeness can be formalized mathematically. Documentation and breadth quantify whether essential data types are present, while density emphasizes longitudinal data capture across encounters. Predictive value links record completeness to downstream inferential strength, bridging data quality and clinical utility. Together, these metrics offer a structured foundation for evaluating EHR completeness beyond qualitative assessment [10,11].

To concretize the four completeness dimensions, we computed RSS on a de-identified MIMIC-IV subset of type 2 diabetes encounters (n = 1000). As shown in Table 2, two medical informaticians selected “required” items for diabetes management; inter-rater agreement was high (κ=0.91). Component values were normalized to [0,1] and aggregated with equal weights:RSS=0.25·Doc+0.25·Bre+0.25·Den+0.25·PV.To enable cross-domain comparability, components were z-normalized within-dataset prior to aggregation.

Table 3 provides an illustrative calculation for three encounters, showing the individual component scores (documentation, breadth, density, and predictive value) and their aggregated RSS values. The visual distribution of these normalized components across encounters is presented in Figure 1, where the bar profiles highlight variation in data completeness among patient records. To ensure cross-domain comparability, each component was z-normalized within its dataset prior to aggregation, and the “required” item set was defined through expert consensus.

### 1.3. Types of Missingness

Rubin’s taxonomy [17] distinguishes between three canonical forms of missingness:Missing Completely at Random (MCAR):(1)P(M∣X,Y)=P(M)

Missingness is independent of both observed and unobserved data.

Missing at Random (MAR):(2)P(M∣X,Y)=P(M∣X)

Missingness depends only on observed data *X*.

Missing Not at Random (MNAR):(3)P(M∣X,Y)≠P(M∣X)

Missingness depends on the unobserved data itself.

Here, *M* is the missingness indicator (M=1 if missing), *X* denotes observed data, and *Y* unobserved values. Identifying the missingness mechanism is crucial for selecting appropriate imputation or modeling strategies in EHR analyses.

As this work offers a conceptual perspective, validation was discussed, rather than empirically performed. We illustrate how simulated data and the Kolmogorov–Smirnov test can support distinguishing random from systematic missingness [18]. Expert interpretation, aligned with prior frameworks [10,12], further informs practical classification. A conceptual illustration summarizing the complementary roles of statistical and expert reasoning in identifying MCAR, MAR, and MNAR patterns is presented in Table 4. This reflective approach emphasizes combining statistical reasoning with expert judgment to guide the robust identification of missingness mechanisms across EHR contexts.

### 1.4. A Taxonomy for EHR Incompleteness

Incompleteness extends beyond data properties to include causal pathways and downstream effects. Figure 2 presents an integrated taxonomy comprising six domains: Data Dimensions (documentation, breadth, density, predictive value); Types of Missingness (MCAR, MAR, MNAR); Causes (human factors, administrative constraints, technological limitations, patient-generated data challenges); Detection and Quantification Approaches (RSS, Kolmogorov–Smirnov test [18], graph theory [19], pattern recognition [20]); Process Gaps (patient, provider, technology, and administrative levels); and Implications (clinical decision risk, interoperability challenges such as HL7/FHIR, regulatory issues, and bias in machine learning models).

This taxonomy underscores that incompleteness is not merely a technical flaw but a systemic process problem with profound clinical, organizational, and ethical consequences.

## 2. Key Research Contributions: Complementary Lenses

Research into EHR incompleteness has progressed through several distinct phases. Early work focused on establishing conceptual frameworks, defining the dimensions of completeness, and highlighting the structural gaps in health records. This conceptual grounding provided the language for later operationalization. Building on these foundations, subsequent studies introduced quantifiable measures that allowed completeness to be evaluated more systematically. Among these, the Record Strength Score (RSS) proposed by Nasir et al. [11] offered a simple yet powerful way to quantify data completeness as the ratio of recorded to required elements within a patient’s clinical context:(4)$$RSS=\frac{N_{\mathrm{present}}}{N_{\mathrm{required}}}$$
where $N_{\mathrm{present}}$ is the number of required elements documented, and $N_{\mathrm{required}}$ is the total expected number of elements. This measure was later extended by Nasir et al. [21] to examine disparities across patient subgroups, demonstrating that incompleteness often reflects underlying inequities in healthcare access and utilization.

As the field matured, more rigorous statistical methods were employed to characterize incompleteness patterns. For instance, Gurupur et al. [22] applied the Kolmogorov–Smirnov (K–S) test [18] to detect whether missingness was random or systematic. The K–S statistic is defined as follows:(5)$$D=\underset{x}{\sup}\left|F_{1}\left(x\right)-F_{2}\left(x\right)\right|$$
where *D* is the maximum distance between the observed empirical cumulative distribution, $F_{1}\left(x\right)$, and the expected distribution, $F_{2}\left(x\right)$. By identifying deviations from expected distributions, this method provides insight into whether incompleteness arises from systematic process gaps, rather than random variation.

More recently, advanced computational approaches have reframed incompleteness as a learnable pattern. Ontology-driven validation has been used to detect structural errors in clinical model design [23], and ML classifiers have been built to predict which physicians will omit documentation in hospital EHR systems [24]. Real-world studies also continue to uncover missingness patterns across demographic, behavioral, and health history fields [25]. These developments emphasize that incompleteness is not merely a technical issue but also a behavioral and systemic one, requiring methods that integrate statistical rigor with contextual understanding. As shown in Table 5, key research contributions on EHR incompleteness are summarized chronologically.

As summarized in Table 6, research on EHR incompleteness has progressed along a chronological trajectory from early conceptual frameworks to advanced computational modeling. Initial work by Weiskopf et al. (2013) [10] established key dimensions of completeness, followed by the introduction of quantitative measures such as the Record Strength Score (RSS) by Nasir and colleagues (2016, 2019) [11,21]. More recent studies have applied graph theory, statistical distribution tests, and machine learning to characterize and predict incompleteness [19,20,22]. Reviews [12,27] and empirical studies [8,28] from 2023 onward reflect both global perspectives and emerging standards, underscoring that incompleteness is a multidimensional issue requiring conceptual, methodological, and policy-driven solutions.

Recent global studies reinforce that incompleteness is not a U.S.-centric phenomenon [12]. Reviews from Europe [12], empirical evaluations in Africa [28], and comparative analyses of EHR versus paper-based records [27] demonstrate that data gaps persist even in advanced health systems and are compounded in resource-limited settings. Taken together, these contributions reveal a multidimensional research trajectory that naturally motivates a broader synthesis of cross-country evidence, which we present in the next section.

While previous studies have examined EHR incompleteness from isolated viewpoints—such as conceptual frameworks [10], completeness metrics like the Record Strength Score (RSS) [11], or methodological bias analyses [12]—our framework provides a unifying perspective that integrates these approaches within a process-oriented model. Specifically, it bridges technical and behavioral dimensions by situating data incompleteness within patient, provider, technology, and policy workflows. This comparison underscores that, unlike earlier studies that primarily focused on measurement or statistical aspects alone, our framework advances the field by linking incompleteness to systemic process gaps and offering actionable pathways for mitigation through multi-level coordination.

## 3. Global Evidence of EHR Incompleteness

While much of the early research on EHR incompleteness originated in the United States, recent work shows that the problem is global in scope. Studies across diverse regions demonstrate that missingness arises not only from technical limitations but also from structural, organizational, and socio-political factors. Table 7 summarizes representative evidence across countries and regions, illustrating that incompleteness is a universal challenge with context-specific manifestations.

As Table 8 demonstrates, incompleteness in EHRs is a universal challenge, but it manifests differently, depending on health system maturity and resources. In high-income countries such as the United States and United Kingdom, incompleteness often arises from workflow limitations, underreporting in structured fields, or a lack of interoperability across vendor systems. In contrast, low- and middle-income countries face structural incompleteness due to limited EHR adoption, partial digitization, and infrastructural challenges such as unreliable connectivity. Hybrid contexts like Latin America highlight the role of policy and governance, with regional variation in reporting standards contributing to data gaps.

Concrete case studies reinforce these findings. In Germany, Wurster et al. (2024) [29] showed that, even after EMR adoption, perioperative and laboratory documentation remained incomplete. In Belgium, Declerck et al. (2024) [30] demonstrated that routine measurements such as height and weight were often missing or inconsistently recorded across hospitals. In the United States, Huang et al. (2023) [4] revealed that incomplete longitudinal data continuity can amplify racial–ethnic disparities in predictive modeling. In Kuwait, AlHussainan et al. (2025) [34] identified critical gaps in safety-related fields across public hospitals. These examples highlight that incompleteness persists even in resource-rich settings and has direct consequences for clinical care, equity, and research validity.

To enable comparability across health systems with differing EHR maturity and data definitions, prior studies highlight the utility of schema alignment through HL7/FHIR mappings and the Observational Medical Outcomes Partnership (OMOP) Common Data Model [15]. Reported incompleteness rates in the reviewed literature were interpreted as relative indicators after accounting for contextual factors such as data model scope and reporting granularity.

Recognizing its global scope is essential for developing mitigation strategies that are context-sensitive while also guided by universal standards. The next section explores the specific process gaps at the patient, provider, technology, and administrative levels, that drive incompleteness within healthcare workflows.

## 4. Process Gaps Driving Incompleteness

As presented in Figure 3 and Table 9, incompleteness in electronic health records emerges from interacting process gaps across four domains: patients, providers, technology, and administrative/policy environments. Recent studies demonstrate how each domain contributes distinctive patterns of missingness and fragmentation.

### 4.1. Patient-Related Factors

Patient-level dynamics shape the continuity, access, and completeness of records. For instance, Teeple et al. (2023) found that missing data patterns vary by race and that these disparities degrade clinical prediction fairness and accuracy [36]. Choy et al. (2024) further found that patients face significant barriers in digital health access—including poor connectivity and limited digital literacy—translating into missing engagement data [37]. Conderino et al. (2024) showed that patients with cost-constrained or delayed care experienced systematically worse data reliability, as self-reported conditions were often absent or under-represented in the record [38].

Earlier interviews also revealed human-related contributors such as the non-disclosure of sensitive conditions, fatigue, and breakdowns in human-to-human communication (e.g., language or cultural barriers), particularly in emergency care [39]. Together, these findings underscore that patient-driven gaps reflect both systemic inequities and day-to-day human behavior.

### 4.2. Provider-Related Factors

Provider workflows, burden, and usability remain central to documentation completeness. Wurster et al. (2024) reported that perioperative and laboratory fields remained incomplete in German surgical departments, even after EMR adoption [29]. Sloss et al. (2024) showed that the redesign of templates, workflow optimization, and the use of medical scribes can reduce burdens and improve documentation [40]. Olakotan et al. (2025) found that poor interface usability and confusing navigation led providers to skip or leave fields incomplete [41].

Qualitative evidence echoes these findings: interviews highlighted human error (deliberate and inadvertent), fatigue, and a lack of verification/validation of inputs as persistent bottlenecks [42]. Medication errors arising from missing or incomplete drug histories exemplify how such gaps can have catastrophic consequences [43].

### 4.3. Technology-Related Factors

System design, interoperability, and UX strongly influence which data are captured and which are omitted. Declerck et al. (2024) found that basic variables like height and weight were inconsistently recorded across Belgian hospitals due to system defaults and inconsistent requirements [30]. Olakotan et al. (2025) noted that suboptimal UIs (lag, non-intuitive layouts) drive incomplete entries [41]. Das et al. (2024) proposed the CSSDH ontology to integrate social determinants with continuity of care standards, showing that standardized ontologies can reduce incompleteness [44].

Interviews further revealed interoperability challenges as a major barrier: fragmentation between EHR systems leads to missing critical data during care transitions, limiting the effectiveness of HL7/FHIR frameworks [19,22]. Issues with unreliable storage (e.g., lack of RAID-based redundancy) can also generate incomplete or inaccessible patient data [45,46]. Patient-generated health data, when unstructured or lacking standards, compounds the problem.

### 4.4. Administrative/Policy-Related Factors

Policy and governance shape the incentives and constraints for completeness. AlHussainan et al. (2025) identified safety-critical omissions in Kuwait’s public hospitals linked to weak verification and oversight [34]. Kim et al. (2025) provide a framework for digital health equity that mandates embedding equity across all phases of digital health governance—from planning and procurement to implementation and monitoring—to prevent systemic access and documentation gaps [47]. In parallel, Hu et al. (2024) find that individuals who are uninsured, less educated, or living in rural or deprived areas are significantly more likely to experience lower EHR completeness tied to access-related policy barriers [48].

Earlier analyses revealed that a lack of standardized protocols and weak accountability perpetuate bottlenecks in data collection, leaving critical fields unvalidated or defaulted to inaccurate values [22]. These deficits in policy and verification systems enable systemic bias in records, which then propagates into analytics and decision-support models.

Together, the four domains illustrated in Figure 3 show that EHR incompleteness arises from a combination of patient access and behavior, provider burden and usability, technology design and interoperability, and policy and administrative oversight. Addressing these demands multi-level interventions: equity in access, usability-centered design, the standardization of technology, and coherent policy frameworks.

## 5. Implications of Incompleteness

The consequences of incomplete electronic health records (EHRs) extend beyond data management to directly influence clinical decision-making, operational efficiency, research validity, and ethical governance. Incompleteness introduces risks that permeate multiple levels of the healthcare ecosystem. This section synthesizes the key implications across four domains: clinical, operational and interoperability, research and machine learning, and ethical and policy considerations.

### 5.1. Clinical Implications

From a clinical perspective, incompleteness compromises the accuracy and timeliness of decision-making. Recent evidence shows that information omissions and documentation defects (“EHRrors”) are common and clinically consequential, encompassing data that are incorrectly recorded or never captured when they should have been [49]. Studies also associate EHR design and use with patient-safety outcomes: incomplete or poorly surfaced histories contribute to medical errors and delay appropriate therapy, whereas the better visibility of prior results can reduce unnecessary tests and downstream risk [50,51]. In high-acuity settings such as emergency departments, the cognitive load created by EHR workarounds and fragmented information further elevates the likelihood of missed data, misinterpretation, and treatment delays [52]. Collectively, these findings emphasize that EHR incompleteness is not merely a data-quality nuisance but a direct threat to patient safety.

### 5.2. Operational and Interoperability Implications

Operationally, missing or inconsistently recorded data drive inefficiencies—duplicated diagnostics, repeated imaging, and reimbursement delays—that increase costs without improving outcomes. Empirical evidence shows that, when clinicians can reliably access prior records across facilities, redundant testing decreases, highlighting how completeness supports the streamlining of care pathways [53]. At the systems level, interoperability frameworks (HL7/FHIR) presuppose complete, well-structured data to enable safe exchange; incomplete payloads degrade handoffs and continuity, especially during transitions of care [54]. Real-world implementations underscore this point: large-scale FHIR deployments reveal recurring “pain points” (e.g., missing required elements, vocabulary mismatches) that impair data flow and undermine the value of interoperability investments [55]. Finally, the documentation burden and misaligned EHR workflows remain persistent operational liabilities that correlate with incomplete entries and downstream rework [56].

### 5.3. Research and Machine Learning Implications

Incomplete EHR data compromise the reliability of biomedical research and the performance of AI models trained on routine care data. Bias can originate from systematic gaps in key covariates (e.g., race/ethnicity), with downstream effects on model calibration, subgroup performance, and external validity [36,57]. State-of-the-art surveys of fair machine learning in health repeatedly identify missingness and data incompleteness as primary sources of algorithmic bias and performance disparities [58,59]. Beyond fairness, incomplete data reduce statistical power, complicate causal inference, and increase reliance on unverifiable assumptions in imputation pipelines; consensus guidance stresses that mitigation must be coupled to explicit characterization of the missingness mechanism and subgroup impact [60]. Relatedly, disparities in how clinicians and staff engage with EHR systems across patient populations can themselves induce patterned missingness, further threatening research generalizability [6].

### 5.4. Ethical and Policy Implications

Ethically, EHR incompleteness raises questions of justice, beneficence, and accountability because the harms of missing data are unevenly distributed and can amplify existing disparities in access and outcomes. Reviews on digital health equity argue that governance should explicitly target data completeness and representativeness as prerequisites for equitable care and AI deployment [47,61]. At the policy level, experts emphasize that semantic interoperability and standard alignment are necessary but insufficient without assurances that the underlying records are complete and fit for reuse [62,63]. In AI-enabled workflows, ethical analyses call for institutional accountability to detect, report, and remedy completeness-related bias before models influence care decisions [64,65,66]. Altogether, ethical and policy considerations underscore that completeness is not merely a technical metric; it is foundational to fair, trustworthy, and legally defensible digital health systems.

## 6. Mitigation Strategies and Future Directions

With the conceptual foundations, process gaps, and broad implications of incompleteness having been outlined, it is critical to turn toward solutions. Addressing incompleteness requires a multi-level approach that integrates technical, organizational, policy, and patient-centered interventions. Each of these domains contributes distinct but complementary mechanisms to reduce missingness, harmonize workflows, and build trust in electronic health records (EHRs). This section synthesizes emerging strategies and future directions across these four domains.

### 6.1. Technical Solutions

Technical innovations offer the most direct pathway to mitigating incompleteness in EHRs. Real-time data quality assessment systems that flag missing, inconsistent, or implausible entries at the point of documentation can substantially reduce downstream errors. Recent studies demonstrate that automated completeness checks integrated into EHR workflows improve both data capture and clinical usability [67,68,69]. Beyond rule-based validation, machine learning methods have been developed to predict likely missing values and harmonize heterogeneous inputs across systems. For example, natural language processing pipelines now extract structured information from unstructured clinical notes, partially closing gaps in structured fields [70,71].

Standardization remains equally critical. Despite the wide adoption of HL7/FHIR, the lack of uniform implementation undermines the consistency of captured data. Ongoing initiatives have emphasized maturing FHIR profiles and strengthening mapping between terminologies such as SNOMED CT and LOINC to reduce structural incompleteness during interoperability [15,72]. Finally, technical advances in federated learning and privacy-preserving record linkage are showing promise in integrating fragmented patient records across institutions without compromising security. For example, Kho et al. (2020) used PPRL in the All of Us Research Program to detect care fragmentation across health provider organizations [73]. Similarly, recent frameworks such as FED-EHR (Wani et al. 2025) and reviews of methodological advances in FL (Zhang et al. 2024) demonstrate that federated models can maintain performance while preserving privacy [74,75]. Altogether, these methods help mitigate longitudinal incompleteness by allowing information linkage and learning across distributed datasets.

### 6.2. Organizational Solutions

Organizational strategies are essential to address the human and workflow dimensions of incompleteness. Evidence consistently shows that high documentation burden and poorly aligned workflows are leading contributors to missing or incomplete EHR data. Interventions such as team-based documentation, medical scribes, and task redistribution have been associated with improved completeness and reduced clinician fatigue [76]. Workflow redesign, usability improvements, and reduced click/navigation burden likewise appear to improve documentation completeness and reduce burnout [77,78].

Training and education represent another critical organizational strategy. Studies reveal that clinicians and ancillary staff often lack sufficient training in structured data entry, resulting in incomplete or inconsistent records [56,61]. Structured onboarding, refresher courses, and role-specific training have demonstrated measurable gains in data completeness across hospital units.

Finally, organizational culture plays an important role. Institutions that emphasize quality improvement and data stewardship, supported by leadership commitment and feedback mechanisms, show higher completeness rates compared to those without such frameworks [79]. Equally important is addressing patient-facing workflows, where initiatives such as improving digital literacy support, patient portals, and structured patient-generated health data entry have shown promise in reducing missingness [80,81].

Together, these findings suggest that organizational solutions are indispensable complements to technical interventions, ensuring that completeness is embedded into day-to-day clinical practice rather than treated as an afterthought.

### 6.3. Policy and Governance Solutions

Policy and governance interventions provide the structural backbone for improving EHR completeness across health systems. Unlike technical or organizational efforts that focus on workflows or tools, governance frameworks establish accountability, incentives, and standards that shape long-term data quality. Recent work emphasizes that national and regional governance structures significantly influence data completeness. For example, a systematic review of digital health governance models demonstrated that countries with mandated reporting standards and centralized oversight (e.g., Denmark, Estonia) achieved higher levels of record completeness compared to systems with fragmented governance [13]. Similarly, regulatory policies enforcing the mandatory coding of diagnoses and medications were associated with significant improvements in EHR completeness in France and Canada [82]. Another crucial area involves the alignment of reimbursement incentives with documentation quality. Studies indicate that, when reimbursement policies reward accurate and complete documentation, providers demonstrate higher compliance with structured EHR data entry [83].

Conversely, poorly designed incentives risk promoting “check-box” behavior without meaningful improvements in data quality. Governance frameworks must also address interoperability and equity. A multicountry 2024 study reported that the absence of coordinated regulatory mandates for HL7/FHIR adoption perpetuates fragmentation and missingness, especially in cross-border care [14,15]. At the same time, the failure to incorporate equity-driven governance exacerbates disparities, as underrepresented populations often face higher levels of incomplete data capture [84].

Together, these findings indicate that policy and governance solutions must extend beyond compliance checklists. Effective governance should embed standardization, accountability, financial alignment, and equity considerations as cornerstones of completeness. Without such frameworks, technical and organizational interventions risk remaining piecemeal, unable to address systemic incompleteness across healthcare ecosystems.

### 6.4. Patient-Centered Solutions

Patients play a pivotal role in mitigating EHR incompleteness, especially through engagement, transparency, and the contribution of patient-generated health data (PGHD). A growing body of evidence suggests that, when patients are empowered as active participants in their health data journey, record completeness and accuracy improve substantially.

One promising approach is the use of patient portals and mobile health applications that allow patients to review and supplement their records. A multicenter trial demonstrated that providing patients with structured opportunities to correct medication lists and allergy information via portals significantly increased data completeness while reducing discrepancies during clinical encounters [85,86]. Similarly, mobile apps integrated with EHRs have been shown to improve the reporting of chronic disease management metrics, such as blood glucose and blood pressure, filling critical gaps in longitudinal monitoring [87,88]. Another important dimension is addressing health literacy and digital divide barriers. Studies show that patients with limited digital literacy are less likely to contribute reliable PGHD, perpetuating disparities in data completeness [89,90]. Targeted training programs and simplified user interfaces can help mitigate this issue, ensuring more equitable patient participation in data contribution.

Finally, involving patients in data governance and quality oversight has gained traction as a mechanism for accountability. A 2025 study reported that incorporating patient representatives in data quality review committees improved trust and identified gaps that clinicians and administrators often overlooked [91]. In addition, expanding the integration of PGHD through wearables and remote monitoring devices provides richer datasets, although the challenges of standardization and validation remain [92,93]. Taken together, patient-centered solutions highlight that data completeness is not solely a technical or institutional responsibility. Patients themselves, when supported with the right tools, literacy, and governance frameworks, become key contributors to more accurate and equitable EHR systems.

Table 10 consolidates the wide range of mitigation strategies discussed in this section. Technical solutions address structural and interoperability gaps, while organizational interventions reduce documentation burden and embed data quality into workflows. Policy and governance frameworks provide accountability and standardization, and patient-centered initiatives extend completeness by empowering individuals to contribute and verify their own health data. Together, these strategies emphasize that no single intervention is sufficient; sustained improvements require coordinated action across all four domains.

## 7. Final Remarks

The synthesis presented in this article underscores that electronic health record (EHR) incompleteness is not solely a technical problem but a systemic process challenge that cuts across patient behavior, provider workflows, technology design, and policy environments. Mitigation, therefore, demands multi-level coordination—combining standardization, usability, equity-oriented governance, and patient empowerment. These reflections aim to consolidate the conceptual and practical implications discussed throughout, setting the stage for the Section 8.

### Limitations

As a perspective, this work does not provide a systematic or quantitative evaluation of all available studies on EHR incompleteness. The synthesis is inherently interpretive and relies on selective integration of conceptual, empirical, and policy literature. While efforts were made to ensure broad coverage and balanced representation, some regional or domain-specific nuances may not be fully captured. Future research could build on this framework through systematic reviews or meta-analyses that quantitatively assess completeness patterns and mitigation outcomes across settings.

## 8. Conclusions

This perspective has highlighted that incompleteness in electronic health records (EHRs) is not merely a technical artifact but a systemic process problem, arising at the intersection of patient behaviors, provider workflows, technological limitations, and policy structures. Across sections, we demonstrated how incompleteness compromises clinical decision-making, weakens interoperability, biases research and machine learning models, and raises serious ethical and governance concerns. From an authorial standpoint, our central argument is that EHR incompleteness must be treated as a foundational barrier to the promise of digital health. It is not sufficient to pursue interoperability or advanced analytics without first ensuring that the underlying data are consistently captured, validated, and contextualized. Our synthesis suggests that mitigation requires a multi-level approach: technical advances such as automated quality checks and federated integration; organizational strategies that reduce documentation burden and embed usability; governance models that align incentives and mandate standards; and patient-centered solutions that empower individuals to contribute and verify their health data. While absolute completeness is unattainable, systematic improvement is both feasible and urgent. By embedding completeness into the design of health information systems and aligning it with equity, accountability, and patient safety, we can shift EHRs from fragmented repositories toward reliable, learning health infrastructures. As researchers and practitioners, we see this as an opportunity: addressing incompleteness is not simply a corrective task but a necessary step toward realizing the transformative potential of digital health for safe, equitable, and globally relevant healthcare systems.

## Figures and Tables

**Figure 1 healthcare-13-02900-f001:**
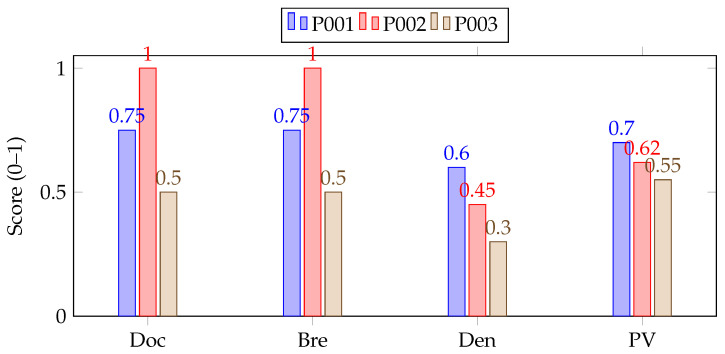
Component profiles for three example encounters (normalized). Equal-weight aggregation yields RSS=0.25(Doc+Bre+Den+PV).

**Figure 2 healthcare-13-02900-f002:**
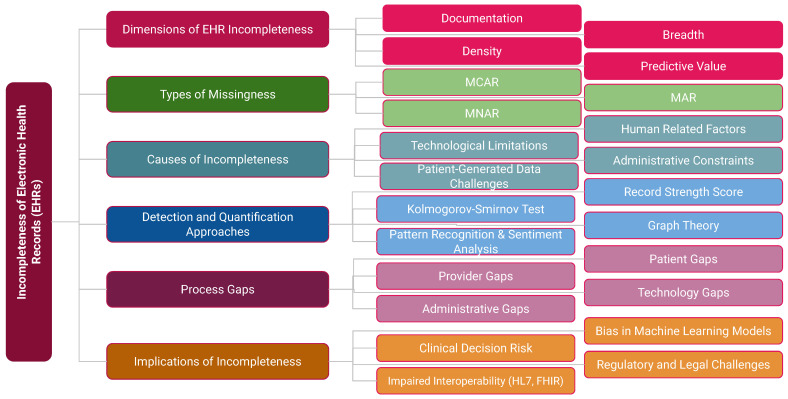
Integrated taxonomy of factors contributing to incompleteness in EHRs, spanning data dimensions, missingness types, causes, detection and quantification methods, process gaps, and implications.

**Figure 3 healthcare-13-02900-f003:**
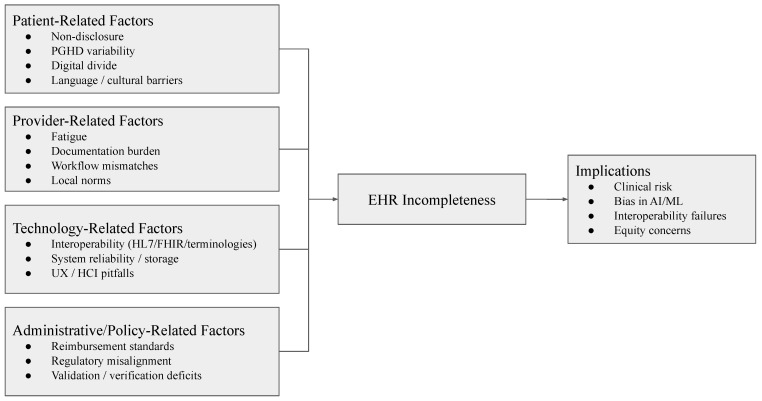
Conceptual model of interacting process gaps (patient, provider, technology, and administrative/policy) leading to EHR incompleteness.

**Table 1 healthcare-13-02900-t001:** Mathematical representations of EHR completeness dimensions [10,11].

Dimension	Formal Expression
Documentation	$D_{i}=\left\{\begin{array}{cc} 1 & \mathrm{if}\;\mathrm{element}\;i\;\mathrm{is}\;\mathrm{documented}\\ 0 & \mathrm{otherwise} \end{array}\right.$, where *i* indexes required elements.
Breadth	$B=\frac{\sum_{i=1}^{k}D_{i}}{k}$, where *k* is the total number of required data types.
Density	$Den=\frac{\sum_{j=1}^{t}E_{j}}{t}$, where $E_{j}$ is the number of entries at observation point *j*, and *t* is the total number of periods.
Predictive Value	$PV=f(\hat{Y}\left(X\right),Y)$, where $\hat{Y}\left(X\right)$ is the predicted outcome based on record *X*, and *Y* is the true outcome (measured by AUROC, accuracy, or similar).

**Table 2 healthcare-13-02900-t002:** Required items and decision rules for the diabetes example (consensus panel of two informaticians).

Required Item	Decision Rule (per Encounter)	Clinical Rationale
HbA1c	≥1 structured HbA1c result recorded	Glycemic control indicator for T2D management.
Systolic/Diastolic BP	≥1 structured SBP/DBP pair recorded	Hypertension risk assessment and therapy titration.
ICD-10 E11.xx Comorbidity	Any E11.xx code present	Confirms T2D diagnosis context and supports classification consistency.
Medication Adherence Note	Any adherence entry (structured or coded NLP)	Indicates therapy effectiveness and disease progression monitoring.

**Table 3 healthcare-13-02900-t003:** Example component scores for three encounters (normalized to [0,1]). Den = entries per 90 days; PV = standardized feature importance in 6-month readmission model.

Encounter	Doc	Bre	Den	PV	RSS Calculation
P001	0.75	0.75	0.60	0.70	0.25(0.75+0.75+0.60+0.70)=0.70
P002	1.00	1.00	0.45	0.62	0.25(1.00+1.00+0.45+0.62)=0.77
P003	0.50	0.50	0.30	0.55	0.25(0.50+0.50+0.30+0.55)=0.46
Mean ± SD	0.75 ± 0.25	0.75 ± 0.25	0.45 ± 0.15	0.62 ± 0.08	0.64 ± 0.16

**Table 4 healthcare-13-02900-t004:** Conceptual illustration of missingness mechanism identification using statistical and expert approaches.

Mechanism	Statistical Cue (K-S Test Outcome)	Interpretive Insight (Expert Reasoning)
MCAR	Distributions of observed vs. missing values show no significant difference (p>0.05)	Likely random omission unrelated to clinical or system factors.
MAR	Deviations linked to observed covariates (e.g., age, visit type)	Missingness associated with measurable variables in workflow or population.
MNAR	K-S test reveals significant skew; pattern persists after conditioning (p<0.05)	Data absence related to unrecorded or sensitive factors (e.g., diagnosis disclosure).

**Table 5 healthcare-13-02900-t005:** Representative research contributions on EHR incompleteness, arranged chronologically.

Study/Line of Work	Core Contribution	Limitations/Notes
Weiskopf et al. (2013) [10]	Defined four dimensions of completeness (documentation, breadth, density, predictive value); provided a conceptual framework for measuring completeness.	Primarily conceptual; lacks quantitative tools for operational assessment across large datasets.
Nasir et al. (2016) [11]	Introduced the Record Strength Score (RSS) to mathematically quantify completeness as the ratio of present to required elements.	Context-specific; requires predefined lists of “required” data elements.
Nasir et al. (2019) [21]	Extended RSS to explore disparities in record completeness across demographics; identified higher missingness in vulnerable populations.	Focused on utilization data; does not generalize across all EHR types.
Wilcox et al. (2021) [19]	Classified missingness into MCAR, MAR, MNAR; applied graph theory to model data flows and identify gaps in smart healthcare environments.	Graph-theoretic models require strong assumptions; practical deployment may be limited in heterogeneous EHR systems.
Gurupur et al. (2021) [20]	Applied distributional approaches (Kolmogorov–Smirnov test) and machine learning to predict incompleteness patterns.	Site-specific validation; computational complexity increases with scale.
Gurupur et al. (2022) [26]	Modeled incompleteness as a learnable pattern using ontologies and recurrent neural networks; linked provider sentiment to completeness.	Requires specialized datasets; generalizability remains underexplored.
Tsiampalis & Panagiotakos (2023) [12]	Reviewed methodological issues in EHR-based epidemiological studies, focusing on biases from missingness.	Narrative scope; lacks empirical validation.
Gurupur et al. (2024) [16]	Provided a standards-based narrative review; emphasized the importance of universal definitions and quality benchmarks.	Descriptive; does not provide quantitative validation.
Fraser et al. (2024) [27]	Compared data quality (completeness, accuracy, timeliness) between paper and EHR implementations.	Context-specific to a limited health system; generalization needed.
Kim et al. (2024) [8]	Highlighted challenges and opportunities in EHR-based research, emphasizing data quality and missingness.	Broad review; does not propose formal metrics.
Mugauri et al. (2025) [28]	Examined a decade of EHR implementations in Sub-Saharan Africa, identifying structural causes of incompleteness.	Regional focus; limited generalizability.

**Table 6 healthcare-13-02900-t006:** Representative contributions on EHR incompleteness (approach, contribution, and typical limitations).

Study/Line of Work	Core Contribution	Typical Limitations/Notes
Conceptual frameworks	Definitions; dimensions of completeness; quality measures	May lack operational metrics; variable generalizability
Graph/flow modeling & missingness classes	Characterizes data sources and flows; clarifies MAR/MCAR/MNAR	Requires assumptions; limited by observability of processes
Record Strength Score (RSS)	Quantifies completeness relative to a required set	Dependent on context-specific “required” element lists
Distributional testing/ML	Detects systematic patterns; links to sentiments/workflows	Needs robust validation; sensitive to site-specific idiosyncrasies
Standards/benchmarks	Shared definitions, reporting, and evaluation guidance	Adoption/implementation gaps across vendors/sites

**Table 7 healthcare-13-02900-t007:** Cross-country evidence of EHR incompleteness.

Region/Country	Context/Dataset	Key Incompleteness Patterns	Reference(s)
United States	Medicare, Medicaid, multi-institutional EHRs	Under-recording of chronic conditions, incomplete problem lists, disparities across demographics.	[10,21]
United States	Multi-institutional predictive modeling	Data discontinuity amplifies racial-ethnic disparities; incomplete longitudinal capture reduces fairness.	[7]
United Kingdom	NHS general practice and hospital records	Missing ethnicity, lifestyle factors (e.g., smoking), and mental health severity scores.	[12]
Nordics (Sweden, Denmark)	National health registries	Underreporting of comorbidities, registry update delays, incomplete linkage across systems.	[12]
Germany	Surgical departments post-EMR adoption	Some documentation fields improved, but perioperative and lab data remained incomplete.	[29]
Belgium	Seven-hospital case study	Basic measures (height, weight) inconsistently recorded across sites and specialties.	[30]
China	Urban vs. rural hospitals	Documentation disparities; fragmented records across health tiers.	[31]
India	Mixed public–private settings	Low EHR adoption; structural incompleteness due to partial digitization.	[32]
South Korea/Japan	Advanced hospital EHRs	Missing behavioral and lifestyle data; limited PGHD integration.	[33]
Kuwait	Public hospitals	Quantitative evaluation of EHR safety revealed substantial missing safety-related fields.	[34]
Sub-Saharan Africa	HIV/maternal health programs	Missing labs, imaging, and continuity data due to infrastructure constraints.	[28]
Latin America (Brazil, Mexico)	National/state health systems	Inconsistent reporting; missing vaccination and chronic disease data.	[35]

**Table 8 healthcare-13-02900-t008:** Cross-country evidence of EHR incompleteness (illustrative).

Country/Region	Context/Dataset	Salient Incompleteness Pattern	Reported/Expected Impact
United States	Claims + EHR hybrids	Under-recorded chronic condition severity; problem list gaps	Misclassification; reimbursement issues; model bias
United Kingdom	GP/EHR registries	Missing ethnicity, smoking, mental health fields	Limits epidemiology; equity analyses
Nordics (SE/DK)	National registries	Underreporting of comorbidities; lag in updates	Risk adjustment drift; outcome misestimation
China	Mixed hospital systems	Rural/urban documentation disparities	Fragmented longitudinal care; biased analytics
India	Low EHR penetration	Structural incompleteness; paper-to-digital transitions	Limited secondary use; safety risks
Sub-Saharan Africa	Pilots/verticals	Missing labs/imaging; connectivity/storage constraints	Clinical delays; surveillance blind spots
Latin America (BR/MX)	State/regional systems	Inconsistent reporting across jurisdictions	Program evaluation bias; data linkage failures

**Table 9 healthcare-13-02900-t009:** Summary of key process gaps contributing to EHR incompleteness.

Domain	Representative Findings (Cross-Country Evidence)	Key Contributing Factors and Gaps
**Patient Level**	Missing data correlate with demographic, socioeconomic, and access disparities (U.S., UK, Africa).	Non-disclosure of sensitive information; variability in patient-generated health data (PGHD); digital divide; language and cultural barriers.
**Provider Level**	Incomplete documentation and workflow burden prevalent in Europe and the U.S.	Clinician fatigue; high documentation load; workflow misalignment; insufficient validation and feedback mechanisms.
**Technology Level**	Interoperability and usability gaps in Belgium, Germany, and Asia impact completeness.	Poor interface usability (UI/UX); inconsistent data standards (HL7/FHIR); fragmented storage; limited PGHD integration.
**Policy/Administrative Level**	Governance and regulatory variation shape documentation quality (Kuwait, Canada, Denmark).	Lack of standardized protocols; weak accountability; misaligned incentives; limited enforcement of documentation quality.

**Table 10 healthcare-13-02900-t010:** Summary of mitigation strategies for addressing EHR incompleteness across technical, organizational, policy, and patient-centered domains.

Domain	Example Strategies	Key Notes/Limitations
Technical Solutions	- Real-time data quality checks. - NLP extraction from clinical notes. - Federated learning for fragmented data. - Enhanced FHIR/SNOMED/LOINC mapping.	Effective for structural gaps but require robust IT infrastructure and standardization across vendors.
Organizational Solutions	- Team-based documentation. - Medical scribes. - Structured training programs. - Workflow redesign with clinician input.	Reduce documentation burden and improve completeness, but sustainability depends on culture and leadership support.
Policy and Governance	- National reporting standards. - Regulatory incentives for complete coding. - Alignment of reimbursement with quality. - Equity-focused governance models	Provide long-term accountability but risk “checkbox compliance” if not paired with monitoring and equity safeguards.
Patient-Centered Solutions	- Patient portals with correction rights. - Mobile health apps for PGHD. - Health literacy interventions. - Patient involvement in quality oversight.	Improve completeness and trust, but effectiveness depends on digital literacy, access, and the usability of tools.

## Data Availability

No new data were created or analyzed in this study. Data sharing is not applicable to this article.

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
