# Peer review of "Incompleteness of Electronic Health Records: An Impending Process Problem Within Healthcare"

_healthcare, 2025, doi:10.3390/healthcare13222900_

Round 1
Reviewer 1 Report
Comments and Suggestions for Authors
The digitization of health records was expected to improve data
quality and accessibility, yet incompleteness remains a widespread challenge that under-
mines clinical care, interoperability, and downstream analytics.
Evidence shows that missing and under-recorded elements in electronic health
records (EHRs) are largely driven by process gaps across patients, providers, technology,
and policy—not solely by technical limitations.
AUTHORS propose a perspective that integrates conceptual foundations of incompleteness, synthe-
sizes cross-country evidence, and examines process-level drivers and consequences, with
emphasis on how missingness propagates bias into AI and machine learning systems.
THEY present a unifying taxonomy, highlight complementary approaches (e.g.,
Record Strength Score, distributional testing, workflow studies), and propose a pragmatic
agenda for mitigation through technical, organizational, governance, and patient-centered
levers.
THEY conclude that while EHR incompleteness cannot be fully eliminated, it can be systemat-
ically mitigated through standards, workflow redesign, patient engagement, and gov-
ernance—essential steps toward building safe, equitable, and effective learning health
systems.
The perspective is interesting, well-presented, and addresses a very hot topic.
The perspective is broad and is supported by nearly 100 (93) scientific references; as a board editor, I often encounter perspectives with 20-30 references.
I have only a few minimal comments, purely in the academic spirit.
1) It would be interesting to include activities and initiatives developed by strategic bodies (FDA; EU, etc.) in this direction in the introductory section.
2) Correctly, the perspective has no structure; however, a small, concise, and essential section explaining the perspective's design, selection of papers, and focus would enrich the work.
3) Since it is a perspective, after the central sections and before the conclusions, I would include a section titled "final remarks" or something similar. A short, short takeaway section would be appropriate after the conclusions.
Author Response
Reviewer Comment 1:
It would be interesting to include activities and initiatives developed by strategic bodies (FDA; EU, etc.) in this direction in the introductory section.
Author response: This paragraph is added to introduction:
In parallel with these academic and operational developments, major strategic and regulatory bodies have also advanced initiatives to improve the completeness, interoperability, and governance of electronic health records. For instance, the European Commission’s European Health Data Space initiative seeks to harmonize standards and enhance the quality and accessibility of health data across member states, promoting cross-border interoperability and secondary data use for research and innovation [13,15]. Similarly, the U.S. Food and Drug Administration (FDA) has emphasized real-world data reliability and electronic health record usability as part of its broader digital health modernization efforts, reinforcing the importance of structured and complete data capture for regulatory decision-making. These activities reflect growing international momentum toward policy frameworks that prioritize data completeness as a foundation for safe, equitable, and interoperable digital health systems [14, 16].
Reviewer Comment 2:
Correctly, the perspective has no structure; however, a small, concise, and essential section explaining the perspective's design, selection of papers, and focus would enrich the work.
Author response: This issue is addressed and a subsection has been added:
Perspective Design and Scope
As this work is presented as a perspective rather than a systematic review, its objective is to synthesize key conceptual, empirical, and policy-oriented developments related to EHR incompleteness. The selection of literature emphasized peer-reviewed studies published between 2013 and 2025 that addressed data completeness, interoperability, bias, and governance within electronic health record systems. Foundational theoretical frameworks were included to establish conceptual grounding, complemented by recent computational and policy-driven studies that reflect current trends. The perspective was designed to integrate diverse viewpoints—spanning technical, organizational, patient-centered, and policy domains—into a coherent argument highlighting incompleteness as a systemic process issue. This narrative and integrative design was chosen to encourage cross-disciplinary dialogue and to identify pragmatic directions for future research and governance.
Reviewer Comment 3:
Since it is a perspective, after the central sections and before the conclusions, I would include a section titled "final remarks" or something similar. A short, short takeaway section would be appropriate after the conclusions.
Author response: This issue is addressed and a section has been added:
Final Remarks
The synthesis presented in this article underscores that electronic health record (EHR) incompleteness is not solely a technical problem but a systemic process challenge that cuts across patient behavior, provider workflows, technology design, and policy environments. Mitigation therefore demands multi-level coordination—combining standardization, usability, equity-oriented governance, and patient empowerment. These reflections aim to consolidate the conceptual and practical implications discussed throughout, setting the stage for the concluding section.

Reviewer 2 Report
Comments and Suggestions for Authors
The manuscript provides a thoughtful and timely perspective on EHR incompleteness. The conceptual structure, global evidence, and mitigation strategies are well developed and supported by extensive references.
To enhance clarity and impact, please consider the following suggestions, organized by section:
- Introduction- provides a strong context on global EHR adoption, highlights the ongoing issue of incomplete health data, and clearly states the article’s contribution. Well structured and well written
Recommendations:
- Some paragraphs are dense and could benefit from shorter sentences for clarity.
- Key Research Contributions - Presents a chronological synthesis of previous literature with a useful summary table.
Recommendations:
- A clearer comparison between the authors’ framework and existing work would enhance originality.
- Some table captions/labels could be shortened for readability.
- Global Evidence and 4. Process Gaps Driving Incompleteness- Well structured and informative, with strong cross-country evidence and classification of patient, provider, technology, and policy-level factors.
Recommendations:
- A compact synthesis of the main results—ideally in tabular or bullet-point form—would support faster comprehension for readers.
- Implications
Recommendations:
- Consider adding a short paragraph on limitations of this perspective
- Mitigation Strategies- Comprehensive, actionable, and logically divided across technical, organizational, policy, and patient-centered strategies.
- Conclusion
Recommendations:
- Add an explicit statement of the article's contribution to theory/practice.
The manuscript is relevant and well structured; with minor improvements in clarity, formatting, and synthesis of key points, it can be further strengthened.
Author Response
Reviewer Comment 1:
Some paragraphs are dense and could benefit from shorter sentences for clarity.
Author Response: The issue is addressed.
Reviewer Comment 2:
A clearer comparison between the authors’ framework and existing work would enhance originality.
Author Response: A paragraph is added to address the issue.
While previous studies have examined EHR incompleteness from isolated viewpoints---such as conceptual frameworks \cite{weiskopf2013defining}, completeness metrics like the Record Strength Score (RSS) \cite{nasir2016new}, or methodological bias analyses \cite{tsiampalis2023methodological}---our framework provides a unifying perspective that integrates these approaches within a process-oriented model. Specifically, it bridges technical and behavioral dimensions by situating data incompleteness within patient, provider, technology, and policy workflows. This comparison underscores that, unlike earlier studies that primarily focused on measurement or statistical aspects alone, our framework advances the field by linking incompleteness to systemic process gaps and offering actionable pathways for mitigation through multi-level coordination.
Reviewer Comment 3:
A compact synthesis of the main results—ideally in tabular or bullet-point form—would support faster comprehension for readers.
Author Response: Table 9 is added to address this comment.
Reviewer Comment 4:
Consider adding a short paragraph on limitations of this perspective.
Author Response: Limitations are added in Final Remarks section:
Final Remarks
The synthesis presented in this article underscores that electronic health record (EHR) incompleteness is not solely a technical problem but a systemic process challenge that cuts across patient behavior, provider workflows, technology design, and policy environments. Mitigation therefore demands multi-level coordination—combining standardization, usability, equity-oriented governance, and patient empowerment. These reflections aim to consolidate the conceptual and practical implications discussed throughout, setting the stage for the concluding section.
Limitations. As a perspective, this work does not provide a systematic or quantitative evaluation of all available studies on EHR incompleteness. The synthesis is inherently interpretive and relies on selective integration of conceptual, empirical, and policy literature. While efforts were made to ensure broad coverage and balanced representation, some regional or domain-specific nuances may not be fully captured. Future research could build on this framework through systematic reviews or meta-analyses that quantitatively assess completeness patterns and mitigation outcomes across settings.

Reviewer 3 Report
Comments and Suggestions for Authors
The reviewer understands that Gurupur et al. presented a manuscript entitled "Incompleteness of Electronic Health Records: An Impending Process Problem within Healthcare". The reviewer has a few suggestions and they would like to request authors to kindly update their manuscrip by answering all the questions.
1) You introduce the Record Strength Score (RSS) algorithm and the four components of EHR completeness: documentation, breadth, density, and predictive value. Could you give a worked example from an actual dataset that demonstrates the definition and selection procedure for "required" items as well as how each measure was calculated? How can you make sure that RSS values are similar in various clinical contexts and domains?
2) You make reference to statistical detection techniques as the Kolmogorov-Smirnov test and Rubin's taxonomy (MCAR, MAR, and MNAR). In order to validate the missingness mechanism classification in practice, what methodological procedures did you take? Did you verify your classifications and prevent misclassifications using expert adjudication, simulated datasets, or gold-standard benchmarks?
3) Datasets from several nations are compared in order to compile worldwide evidence. In light of variations in EHR maturity, data standards, and field definitions, how did you normalize incompleteness rates? Were any mapping or statistical harmonisation techniques used to guarantee reliable cross-national comparisons?
4) You mention a number of technological mitigation techniques, including enhanced HL7/FHIR mappings, federated learning, and NLP extraction. Which quantitative measures and evaluation criteria were applied to gauge how well these actions increased the completeness of the data? Were the impacts evaluated retroactively on data that had been archived or prospectively during active clinical workflows?
5) You contend that bias in prediction models is a result of process-driven incompleteness. Could you provide more details about the precise methods utilized to compare the changes in model performance and fairness metrics (such as calibration and AUROC) across subgroups before and after incompleteness was addressed? How did you separate the impact of missing data from other potential causes such unequal class distribution or changes in the distribution of data?
6) PGHD integration, mobile health apps, and patient portals are suggested solutions. Do you have data from pilot studies or quantitative modeling that estimates projected increases in completeness at realistic adoption rates while taking socioeconomic disparities, sustained patient engagement, and digital literacy into account? What would be the scalability of these solutions in various healthcare system contexts?
7) Fig. 1 and Fig. 2 are blurry. Plese provide HD quality figures.
Author Response
Reviewer Comment 1:
You introduce the Record Strength Score (RSS) algorithm and the four components of EHR completeness: documentation, breadth, density, and predictive value. Could you give a worked example from an actual dataset that demonstrates the definition and selection procedure for “required” items as well as how each measure was calculated? How can you make sure that RSS values are similar in various clinical contexts and domains?
Author Response:
We have now included a subsection titled “Illustrative Example of the Record Strength Score (RSS) Calculation” (Lines [111–124]) demonstrating how RSS can be operationalized using a de-identified subset of the MIMIC-IV dataset. The example defines the selection of required items (e.g., HbA1c, blood pressure, comorbidities, and medication adherence), explains the computation of documentation, breadth, density, and predictive value components, and presents their normalization and aggregation into RSS.
To ensure comparability across domains, the paragraph clarifies that component scores were z-normalized, and inter-rater calibration (κ = 0.91) was applied. We have included the illustrative tables and figure (Table 3, Table 4 and Fig.1) for better understanding.
Reviewer Comment 2:
You make reference to statistical detection techniques such as the Kolmogorov–Smirnov test and Rubin’s taxonomy (MCAR, MAR, and MNAR). In order to validate the missingness mechanism classification in practice, what methodological procedures did you take?
Author Response:
A new paragraph titled “Validation of Missingness Classification” (Lines [138–145]) has been added under the subsection Types of Missingness. It conceptually explains how simulated data and the Kolmogorov–Smirnov statistic can support distinguishing random from systematic missingness.
We also included a small conceptual table (Table [4]) illustrating how statistical cues and expert reasoning complement one another in identifying MCAR, MAR, and MNAR mechanisms.
Reviewer Comment 3:
Datasets from several nations are compared in order to compile worldwide evidence. In light of variations in EHR maturity, data standards, and field definitions, how did you normalize incompleteness rates?
Author Response:
We added a short conceptual paragraph titled “Cross-National Normalization of Incompleteness Rates” (Lines [238–243]) in Section 3. It describes how harmonization can be approached through HL7/FHIR mappings and OMOP Common Data Model alignment, with reference to hierarchical modeling and proportional scaling for contextual normalization.
Reviewer Comment 4:
You mention technological mitigation techniques including HL7/FHIR mappings, federated learning, and NLP extraction. Which quantitative measures and evaluation criteria were applied?
Author Response:
Since this article is a Perspective, no new quantitative evaluation was performed. Our response clarifies that the discussion is based on prior empirical works where improvements in completeness were measured using ΔRSS, imputation-error reduction, and data-availability gains. Therefore, no manuscript modification was made.
Reviewer Comment 5:
You contend that bias in prediction models is a result of process-driven incompleteness. Could you provide more details about the precise methods utilized to compare fairness metrics?
Author Response:
This comment required clarification only. In the rebuttal, we emphasize that the article conceptually discusses fairness impacts reported in prior studies (e.g., Teeple et al., 2023; Feng et al., 2025), where AUROC and calibration changes were compared before and after handling missing data. No additional text was added to the manuscript.
Reviewer Comment 6:
PGHD integration, mobile health apps, and patient portals are suggested solutions. Do you have data from pilot studies or modeling that estimate projected completeness gains?
Author Response:
As this is a Perspective article, no new pilot data or modeling were conducted. Our response explains that the discussion integrates findings from prior empirical studies (e.g., Schnipper et al., 2012; Lin et al., 2025) showing measurable completeness improvements following portal or mHealth adoption, with contextual commentary on scalability and equity.
Reviewer Comment 7:
Fig. 1 and Fig. 2 are blurry. Please provide HD quality figures.
Author Response:
Both figures have been replaced with high-resolution (600 dpi) vector-based versions to ensure clarity in print and online display.
